# New Challenges in the Diagnosis of Kidney Damage Due to Immune Checkpoint Inhibitors Therapy: An Observational Clinical Study

**DOI:** 10.3390/diagnostics13152524

**Published:** 2023-07-28

**Authors:** Laura Vicente-Vicente, Alfredo G. Casanova, Javier Tascón, Marta Prieto, Ana I. Morales

**Affiliations:** 1Toxicology Unit, Universidad de Salamanca, 37007 Salamanca, Spain; lauravicente@usal.es (L.V.-V.); alfredogcp@usal.es (A.G.C.); javiertascon_96@usal.es (J.T.); martapv@usal.es (M.P.); 2Institute of Biomedical Research of Salamanca (IBSAL), 37007 Salamanca, Spain; 3Group of Translational Research on Renal and Cardiovascular Diseases (TRECARD), Universidad de Salamanca, 37007 Salamanca, Spain; 4RICORS2040-Instituto de Salud Carlos III, 28029 Madrid, Spain

**Keywords:** immune checkpoint inhibitors, nephrotoxicity, diagnostic, biomarkers

## Abstract

In recent years, immunotherapy has been postulated as one of the most effective strategies in the fight against cancer. The greatest success in this field has been achieved with the inhibition of molecules involved in slowing down the adaptive immune response by immune checkpoint inhibitors (ICIs). Despite its efficacy, ICI treatment has side effects. Regarding kidney damage, it is estimated that 4.9% of patients treated with ICIs develop renal injury. Furthermore, cancer patients who develop renal dysfunction have a worse prognosis. Current diagnostics are insufficient to predict the underlying renal injury and to identify the type of damage. Our hypothesis is that the renal injury could be subclinical, so the possibility of using new urinary biomarkers could be a useful diagnostic tool that would allow these patients to be managed in a preventive (risk biomarkers) and early (early biomarkers) way and even to clarify whether the renal damage is due to this therapy or to other factors (differential diagnostic biomarkers). A prospective study to validate risk and early and differential biomarkers in patients treated with ICIs is proposed to test this hypothesis. The results derived from this study will improve the clinical practice of cancer treatment with ICIs and therefore the life expectancy and quality of life of patients. Trial Registration: ClinicalTrials.gov, NCT04902846.

## 1. Background

Advances in the knowledge of the immune response against tumours, coupled with an understanding of mechanisms of the evasion of these, have led to the development of novel, immune response-focussed treatments. The most promising immune response-focussed treatment is the inhibition of molecules involved in breaking the adaptive immune response. Examples include cytotoxic T-lymphocyte antigen-4 (CTLA-4), and programmed cell death protein-1 (PD-1), with their respective ligands. Compounds capable of blocking the action of these molecules are called immune checkpoint inhibitors (ICIs) [1].

Owing to their mechanism of action, ICI use is limited due to their association with adverse effects derived from an autoimmune response [2]. The most frequently involved organs include the liver, lungs, skin, gastrointestinal tract, and endocrine system [3,4]. Although nephrotoxicity is less common, its effects worsen the prognosis of cancer patients [5]. 

Therapies based on combinations of ICIs have recently been developed to avoid acquired tumour treatment resistance. The use of ICIs in combination with chemotherapy has been shown to improve treatment response. However, this also increases the risk of nephrotoxic adverse events [6]. Clinical trials report the incidence of nephrotoxic adverse events during ICI therapy to rise from 2% when administered as monotherapy to 4.9% when administered in combination [7,8]. Some studies suggest that this increased incidence is even higher [5,9,10]. The latency time between the initiation of ICI therapy and the development of kidney damage is observed to range from 13 to 40 weeks, which suggests that the ICIs might have a secondary mechanism that causes the injury [11]. However, the mechanisms responsible for kidney damage are unknown. One of the most widely proposed hypotheses relates to the development of autoimmunity induced by ICIs. In the kidney, this could be triggered against a potential antigen on the cell surface of proximal tubular epithelial cells, causing them to act as pseudo-antigen-presenting cells [12]. Moreover, electrolyte disorders, including abnormal blood calcium, magnesium, potassium, phosphate, and Fanconi syndrome, are all also suggested to be involved in kidney damage [13].

In most cases, kidney damage caused by ICIs is acute tubulointerstitial nephritis (ATIN). However, other types of damage include immunoglobulin A (IgA) nephropathy, membranous nephropathy, lupus-like immune complex glomerulonephritis, C3 glomerulopathy, focal segmental glomerulosclerosis, thrombotic microangiopathy, and pauci-immune crescentic glomerulonephritis [14,15]. In some cases, nephrotoxic effects lead to treatment withdrawal, owing to the risk of permanent and irreversible kidney damage [16]. However, this withdrawal may not be appropriate if nephrotoxicity does not progress to permanent kidney damage. As such, improvement in the diagnosis of kidney damage associated with ICI treatment is paramount. 

Strategies that focus on predicting the likely progression of kidney damage (i.e., artificial intelligence [17]), would enable the most effective pharmacological strategy against the tumour to be implemented without causing further kidney injury. Current clinical and laboratory analyses for the diagnosis of kidney damage are inadequate due to current serum creatinine or urea biomarkers only indicating established damage [18]. As such, for accurate diagnosis, a kidney biopsy is usually required [7]. This is not ideal, being invasive and posing a risk to the patient [4]. Despite this, in many cases, a biopsy is necessary to diagnostically confirm ATIN, which can be treated with corticosteroids [19]. However, ICI therapy should be withdrawn if a patient progresses to advanced stages of AKI (stage 2 or 3) according to the KDIGO categorization [13,20]. For these reasons, effective clinical diagnoses and biomarkers are currently being redefined to detect and differentiate renal involvement in its different stages [21]. 

Patient predisposition to and risk of kidney damage should ideally be assessed prior to the commencement of ICI treatment. This would help identify and stratify patients according to their susceptibility to kidney damage. Susceptibility biomarkers have been identified in preclinical models, and their usefulness is currently being evaluated in different clinical settings [22,23,24,25,26].

A new, early diagnosis that predicts elevated creatinine levels is currently being studied [21]. This would enable the detection of kidney damage at the very earliest of stages. Potential early diagnostic biomarkers derived from different stages of kidney damage are N-acetyl-beta-D-glucosaminidase (NAG), kidney injury molecule-1 (KIM-1), and neutrophil gelatinase-associated lipocalin (NGAL).

Furthermore, several authors [27,28,29] have all sought to establish differential diagnostic techniques to distinguish between different types of kidney damage. Due to the pathology of cancer and the prescription of different drugs to cancer patients, a range of kidney injuries can occur. These include ischaemic and/or nephrotoxic tubular injury, paraneoplastic kidney injury, glomerular injuries, and tubular obstruction. All possible causes of damage must be carefully evaluated to ensure a correct diagnosis prior to undergoing ICI treatment. The characteristic pathophysiological processes of each type of kidney injury can give rise to differing degrees of damage. As such, a biomarker that can differentially indicate the type of damage would be invaluable and help reduce the need for invasive procedures, such as biopsy.

This study proposal has huge clinical relevance. Knowledge of novel biomarkers would enable the stratification of treatment, thus helping to avoid and minimise kidney damage at different treatment stages. Risk biomarkers would enable the prediction of individual patient risk. Furthermore, during treatment, early biomarkers would enable the detection of early lesions, and differential diagnostic biomarkers would enable the identification of the type of kidney damage, once detected.

## 2. Study Objectives and Endpoints

The main objective of this study is to validate biomarkers as diagnostic tools for kidney damage in ICI-treated patients, and moreover, whether biomarkers can also be a valid management tool in these patients. 

For this purpose, the following stages associated with kidney damage are proposed (Figure 1): Pre-treatment stage: in this stage, risk biomarkers would predict individual patient risk.Phase I (1st stage): this stage is during the first stages of treatment; early biomarkers would detect early lesions.Phase II (2nd stage): this stage is during treatment; differential diagnostic biomarkers would help diagnose the type of lesion in established kidney damage.

With these study objectives, we aim to determine whether biomarkers can be a valid tool in the management of kidney damage in patients treated with ICI-based immunotherapy, responding to an unresolved diagnostic need.

**Figure 1 diagnostics-13-02524-f001:**
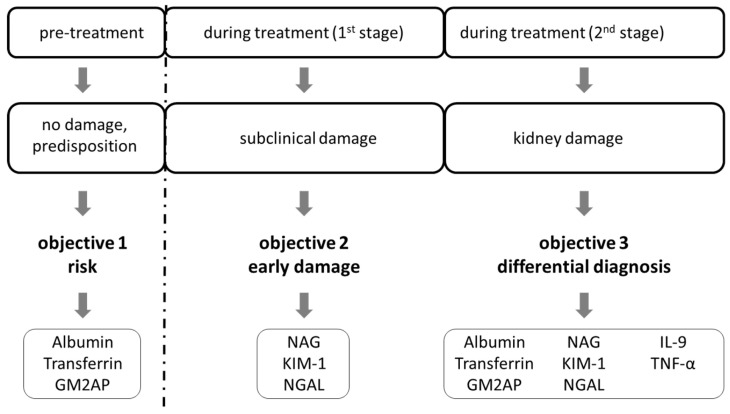
Study design outline. Objectives based on treatment progression. GM2AP: GM2 ganglioside activator protein; NAG: N-acetyl-beta-D-glucosaminidase; KIM: kidney injury molecule-1; NGAL: neutrophil gelatinase-associated lipocalin; IL-9: Interleukin 9, TNF-α: tumour necrosis factor alpha.

## 3. Methods and Design

### 3.1. Study Design

This prospective study will include patients undergoing ICI-based immunotherapy or combined ICI/chemotherapy treatment with platinum compounds. Patients will be recruited from two medical oncology services in Spain, Complejo Asistencial Universitario de Salamanca and Hospital Clínico Universitario de Valladolid. This study will not alter the standard clinical procedures. 

Patients included in the study will be classified into 2 groups. Group 1, patients treated with single or combined immunotherapy, and group 2, patients treated with immunotherapy combined with chemotherapy. The 2 groups will be further divided into 2 subgroups, controls and cases. The cases subgroup will include patients who develop acute kidney damage during their treatment, and controls will include those who do not develop kidney damage during their treatment. Acute kidney damage will be defined by elevated creatinine, according to KDIGO criteria [20], electrolyte alterations, proteinuria, and/or haematuria. 

### 3.2. Ethical Aspects and Patient Information

The researcher will provide all relevant study information to the recruited patients before obtaining their consent. A patient information sheet will be provided in the local language and prepared according to ICH GCP guidelines (ICH Topic E6, 1995). Prior to their inclusion in the study, all participants will sign an informed consent form, as established in the Declaration of Helsinki [30] and WHO standards [31] for observational studies. The study protocol does not modify any standard sanitary procedures. Patients will be informed of the potential benefits of the project and the reasons why it is necessary to collect biological samples. The confidentiality of the study participants will be ensured, complying with current legislation on personal data protection (Law 3/2018 of The Protection of the Official Law on Personal Data). The study will also comply with Act 14/2007, 3rd July, on Biomedical Research. 

### 3.3. Inclusion Criteria

Patients are required to meet the following criteria to participate in the study: have provided informed consent, are ≥18 years old, are awaiting treatment with ICI-based immunotherapy or combined ICI/chemotherapy treatment with platinum compounds, and have normal kidney function prior to treatment. Normal kidney function will be defined according to the margins established by the biochemistry service of each hospital prior to treatment.

### 3.4. Exclusion Criteria

Patients with an existing disease or condition that would interfere with either participation or study evaluations will be excluded. 

### 3.5. Withdrawal of Patients

Patients will be freely able to withdraw from the study at any time without having to provide an explanation. The study subjects will be withdrawn if a clinically relevant event occurs that affects patient safety.

### 3.6. Sample Size

To determine the necessary study sample size, a type I error (α = 0.05), statistical power of 1-β = 0.80, and frequency of case and control occurrence will be used. Since the last parameter cannot yet be determined, a *p*-value of 0.50 was taken. This guarantees the largest sample size. A population size of 186 was obtained. It is necessary to estimate potential losses of patients for various reasons including loss of information, death, and withdrawal. As such, the sample size was increased to (N = n (1/1 − R)), where n is the number of patients without losses and R is the estimated loss (15%). Therefore, the final estimated total sample size is 220 patients, a total of 110 from each hospital. 

### 3.7. Patient Data and Sample Collection

#### 3.7.1. Patient Data

Patient data will be collected using a database that records the following variables: age, sex, family history, personal history, tumour type, antitumour treatment regimen, clinical patient characteristics, comorbidity factors, allergies, prescribed medication, occasional drugs, and follow-up data.

#### 3.7.2. Urine and Blood Sampling

Blood and urine samples will be collected from each patient prior to starting ICI treatment (pre-ICI) and after each ICI treatment (post-ICI)

#### 3.7.3. Processing and Storage of Samples

The biochemistry service of each hospital will process all patient blood samples collected. Urine samples collected at the Complejo Asistencial Universitario de Salamanca will be handled and stored in the internal hospital sample bank. Urine samples collected from the Hospital Clínico Universitario de Valladolid will be handled and stored in the hospital’s internal oncology unit. After collection, urine will be centrifuged and subsequently aliquoted and frozen at −80 °C until use.

#### 3.7.4. Analysis of Plasma Samples

At blood sampling times, the plasma concentrations of creatinine, urea, and electrolytes and the estimated glomerular filtration rate will be analysed and issued by the biochemistry service of the respective hospitals. 

#### 3.7.5. Analysis of Urine Samples

Analytical determinations will be carried out by The Theranostics of Renal and Cardiovascular Diseases Group of the Institute of Biomedical Research of Salamanca (IBSAL). These analyses will include the following.

#### 3.7.6. Urinary Creatinine

Urinary creatinine will be determined using a commercial colourimetric kit based on the Jaffe method [32], “Quantichrom Creatinine Assay Kit” (BioAssay Systems, Hayward, CA, USA). Urinary creatinine is used as a correction factor corresponding to urinary flow; as such, this parameter can be used to establish the biomarker/urinary creatinine ratio for each biomarker.

#### 3.7.7. Proteinuria

Urinary protein excretion will be determined using the Bradford colourimetric method [33].

#### 3.7.8. Risk Biomarkers (Albumin, Transferrin and Gm2 Ganglioside Activator Protein)

For the determination of albumin and transferrin, the commercial kits “Human Albumin ELISA kit” and “Human Transferrin ELISA Kit” (both from Bethyl Laboratories, Montgomery, AL, USA) will be used, respectively. To determine the GM2 ganglioside activator protein (GM2AP), Western blotting will be performed. Briefly, 21 µL of urine from each patient per sampling time will be separated by acrylamide electrophoresis. Proteins will be transferred to an Immobilon-P Transfer Membrane (Millipore, Madrid, Spain) and incubated with the primary antibody against GM2AP (own production). Subsequently, they will be incubated with horseradish peroxidase-conjugated secondary antibody and chemiluminescent detection (Immobilon Western Chemiluminescent HRP Substrate kit; Millipore, Madrid, Spain) will be performed with the ChemiDoc MP imaging system (Bio-Rad, Madrid, Spain).

#### 3.7.9. Early Biomarkers of Kidney Damage (NAG, KIM-1 and NGAL)

The determination of the NAG enzyme activity in urine will be carried out using a commercial colourimetric kit (Diazyme, Poway, CA, USA) following the manufacturer’s instructions. The other biomarkers will be measured using commercial kits based on the ELISA technique: “Human KIM-1 ELISA kit” (Enzo Life Sciences, Lausen, Switzerland) for KIM-1, and the “Human NGAL ELISA Kit 036CE” (BioPorto Diagnostics, Hellerup, Denmark) for NGAL.

#### 3.7.10. Differential Diagnosis 

A comparative statistical study will be carried out to determine whether any of the biomarkers mentioned above are related to different types of kidney damage, specifically, tubular necrosis, glomerular damage, or interstitial nephritis. In addition, two inflammatory biomarkers that can help in this differential diagnosis will be evaluated: Interleukin 9 (IL-9) (Invitrogen IL-9 Human ELISA Kit, Waltham, MA, USA) and tumour necrosis factor alpha (TNF-α) (Invitrogen TNF alpha Human Instant ELISA™ Kit, Waltham, MA, USA).

#### 3.7.11. Statistical Analysis

Data will be analysed using IBM SPSS statistical software, version 25.0 (International Business Machines, Armonk, NY, USA). Different tests will be applied depending on the association to be studied and the type of variable. These will include normality tests (Kolmogorov–Smirnov, Shapiro–Wilk), frequency comparison tests (chi-squared, Fisher’s exact), correlation tests (Pearson, Spearman), comparisons of means for independent data (*t*-test, Mann–Whitney U), ANOVA for repeated measures, and Wilcoxon test for paired data. Finally, multivariate analysis will be used to simultaneously study datasets with several variables for each patient or parameter collected. These multivariate analyses will include regression analysis, general linear models, binary logistic regression, and correspondence analysis. Significance will be defined as a *p*-value ≤0.05. 

Objective 1: Associations between the presence of risk biomarkers (albumin, transferrin, and GM2AP) in urine before treatment and the appearance of kidney damage after treatment will be analysed.Objective 2: The presence of early damage biomarkers will also be evaluated during treatment, comparing the results in the different target groups.Objective 3: Finally, associations will be sought between the studied biomarkers and the etiological diagnosis of kidney damage (tubular necrosis, glomerular damage, or interstitial nephritis).

### 3.8. Dates of the Study

The study was initiated on the 1 January 2021, and the collection of samples is expected to end in 2023. We aim to complete this study in 2023.

### 3.9. Patient and Public Involvement

The results of this study will be communicated to both the participants and the public through a written report. Both social and communication media will be used to disseminate the study findings. The key study findings will be communicated, at the least, in a local congress addressed to individuals from the Salamanca health area. 

A results dissemination conference will be conducted in collaboration with the Spanish Association Against Cancer and The Association of Kidney Patients (ALCER). Here, the relevance of kidney damage in the treatment of cancer patients will be highlighted, along with the importance of detecting kidney damage in its early stages. Moreover, concepts including immunotherapy and biomarkers will be explained to patients.

## 4. Discussion

The diagnosis of kidney damage associated with ICIs is a challenge that must be addressed. The identification of novel biomarkers, which help avoid the requirement for biopsies, is paramount [34]. According to our hypothesis, new predisposition and early and differential biomarkers currently being validated could identify and characterize kidney damage when implemented at the correct ICI treatment time. Such knowledge is crucial for reducing kidney damage complications, the incidence of which is currently increasing as ICI treatment use increases. Kidney damage compromises the life of the patient and their prognosis, since in many cases kidney damage leads to the cessation of ICI treatment.

The study aims to identify predisposition biomarkers, capable of detecting patients susceptible to developing kidney damage prior to starting ICI treatment. Such biomarkers have been defined in preclinical models [22,23,24,25,26]. Clinically, this enables the identification of people lacking renal involvement prior to ICI treatment but who are at a higher risk of developing renal damage during treatment. This could be due to other potentially nephrotoxic treatments, radiological contrasts, and environmental toxins, which do not cause any harm to non-predisposed people. As such, patients can be stratified and managed according to their renal status before treatment. This allows the selection of the most appropriate treatment strategy for each patient. Such biomarkers are currently being validated in different clinical contexts including chemotherapy with cisplatin and the use of contrast media for the diagnosis of cardiac pathologies [22,24]. The studies show that certain biomarker levels are higher in the urine samples of patients susceptible to kidney damage prior to the administration of cisplatin or contrast medium treatment. Such findings clearly indicate the usefulness of the prediction of risk in clinical settings. 

Early diagnosis refers to a new type of biomarkers that anticipate creatinine elevation [21]. As such, this study also aims to improve early kidney damage diagnosis in cases in which ICI treatment is already initiated. In this case, early is defined as a diagnosis made ahead of that by current clinical practice. Existing clinical diagnosis is based on plasma creatinine levels and/or the estimation of glomerular filtration rate. The identification and subsequent detection of the presence of early kidney damage biomarkers could improve the prediction of kidney damage and its course. This would help improve existing treatment protocols through the prevention of the development of kidney damage, which could compromise treatment viability.

Finally, differential diagnosis [27,28,29] is proposed. This study aims to identify biomarkers capable of indicating the different types of kidney damage. This will enable the selection of the most appropriate treatment for each patient and help reduce the need for renal biopsy through a definitive biomarker fingerprint. To achieve this, the study will compare the urinary excretion of biomarkers between patients with kidney damage and those without. As a minimally invasive procedure, this can be conducted at multiple different times over the course of ICI treatment. This would help inform the most appropriate choice of treatment course for each individual patient, maximizing treatment benefits whilst reducing the incidence of unwanted adverse effects. 

## 5. Conclusions

In summary, this study aims to improve the diagnosis of kidney damage associated with ICI treatments. Such knowledge and results obtained may inform the development of novel diagnostic tools for clinicians, improving the accuracy and effectiveness of treatment plans according to individual patient needs. Ultimately, a more stratified diagnostic approach is proposed to help reduce the incidence of kidney damage and improve cancer treatment with ICIs.

## Data Availability

The results of the ICI-TOX study will be disseminated after completion in peer-reviewed journals and presented at conferences.

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
