# Peer review of "New Challenges in the Diagnosis of Kidney Damage Due to Immune Checkpoint Inhibitors Therapy: An Observational Clinical Study"

_diagnostics, 2023, doi:10.3390/diagnostics13152524_

Round 1

Reviewer 1 Report

The study aims to improve the diagnosis of kidney damage associated with ICI treatments, providing novel diagnostic tools for clinicians, improving accuracy and effectiveness of treatment plans according to individual patient needs. The paper is well written and logically organized. Please discuss briefly the role of PD L1 in different clinical settings due to the fundmentale implications related to indications, scores and reporting systems (quote PMID: 35887569). Furthermore Artificial Intelligence has the potential to be a tool in specific context like this (quote PMID: 37333860)

The paper is well written

Author Response

We appreciate the time and efforts by the editor and reviewers in reviewing this manuscript. We are very grateful for their constructive comments to our manuscript “New Challenges in the Diagnosis of Kidney Damage due to Immune Checkpoint Inhibitors Therapy: An Observational Clinical Study”. We hope that the revised manuscript following the reviewers' comments will be suitable for publication in "Diagnosis.

Thanks to the reviewer's suggestion, a sentence has been added in line 70 to include artificial intelligence in the diagnosis of this pathology.

Reviewer 2 Report

Dear Authors:

First of all, as a protocol, the manuscript has done a great job in background explanation, motivation, and reasonable experimental design. The work is in progress and I am sure the results would be valuable. The only lack of significance in the paper is there is no evidence yet, that the protocol could lead to a direct relevance of ICIs and renal injury, and mechanism study of it.

Besides, there are some concerns that need to be mixed.

1. Line 47. The citation is not added to the reference.

2. The statement ' The latency time between the initiation of 50 ICI therapy and the development of kidney damage is observed to range from 13-40 51 weeks, which suggests that ICIs do not exert direct toxicity.' is not correct. Though the time can be long, the damage can still be directly related to the ICIs. You could change to 'which suggests that the ICIs might have secondary mechanism that cause the injury'.

3. '[25–27] have all sought to establish differential diagnostic techniques 91 to distinguish between different types of kidney damage'. You cannot use the citation numbers as the subjects in the sentence. Please revise.

Other than that, I would suggest a minor revision.

Author Response

We appreciate the time and efforts by the editor and reviewers in reviewing this manuscript. We are very grateful for their constructive comments to our manuscript “New Challenges in the Diagnosis of Kidney Damage due to Immune Checkpoint Inhibitors Therapy: An Observational Clinical Study”. We hope that the revised manuscript following the reviewers' comments will be suitable for publication in "Diagnosis.

In response to your suggestions:

The errors observed in the citations of lines 47 and 91 have been corrected, in addition, another similar error has been detected in line 295 that has also been corrected.

The text change suggestion indicated on line 50 has been made